# V2X-UniPool: Unifying Multimodal Perception and Knowledge Reasoning for Autonomous Driving

## Abstract

Autonomous driving (AD) has achieved significant progress, yet single-vehicle perception remains constrained by sensing range and occlusions. Vehicle-to-Everything (V2X) communication addresses these limits by enabling collaboration across vehicles and infrastructure, but it also faces heterogeneity, synchronization, and latency constraints. Language models offer strong knowledge-driven reasoning and decision-making capabilities, but they are not inherently designed to process raw sensor streams and are prone to hallucination. We propose V2X-UniPool, the first framework that unifies V2X perception with language-based reasoning for knowledge-driven AD. It transforms multimodal V2X data into structured, language-based knowledge, organizes it in a time-indexed knowledge pool for temporally consistent reasoning, and employs Retrieval-Augmented Generation (RAG) to ground decisions in real-time context. Experiments on the real-world DAIR-V2X dataset show that V2X-UniPool achieves state-of-the-art planning accuracy and safety while reducing communication cost by more than 80%, achieving the lowest overhead among evaluated methods. These results highlight the promise of bridging V2X perception and language reasoning to advance scalable and trustworthy driving. Our code is available at: `https://anonymous.4open.science/r/V2X-UniPool-7326`.

## 1 Introduction

Autonomous driving (AD) has achieved remarkable progress, however single-vehicle intelligence remains fundamentally constrained by limited sensing range and occlusion. To overcome these challenges, **Vehicle-to-Everything (V2X) communication** has emerged as a promising paradigm, enabling collaborative perception across vehicles and infrastructure (Ignatious et al., 2022; Yoshizawa et al., 2023; Yusuf et al., 2024). By integrating distributed sensing resources, V2X extends perception range and improves safety in complex scenarios (Huang et al., 2023). In parallel, the rapid advancement of language models including **large language models (LLMs)** and **vision–language models (VLMs)** is catalyzing a shift from purely data-driven to knowledge-driven pipelines (Wen et al., 2023). Equipped with common-sense knowledge and reasoning ability, these models offer the potential to support human-like decision-making and more intelligent driving policies in dynamic traffic environments (Yang et al., 2023; Hadi et al., 2024).

Despite their promise, both approaches face fundamental limitations. V2X fusion introduces cross-modal heterogeneity, temporal misalignment, and non-negligible communication overhead (Xu et al., 2022; Yang et al., 2024), hindering the establishment of a unified and temporally consistent representation. Meanwhile, although LLMs and VLMs excel at processing natural language and static images, they cannot directly consume raw multimodal sensor data essential to AD (e.g., LiDAR point clouds, high-frequency trajectories), and are prone to hallucination when deprived of real-time grounding (Sreeram et al., 2024; Bai et al., 2024; Huang et al., 2025). **This exposes a critical gap: existing V2X frameworks remain perception-centric, while language models remain knowledge-centric; no solution unifies the two into a language-based communication paradigm that enables temporally consistent and knowledge-grounded reasoning.**

We propose **V2X-UniPool**, the first framework that bridges V2X perception with language-based reasoning for AD. The system comprises three modules: (i) a translation layer that converts raw

multimodal inputs into a unified language-based representation, (ii) a time-indexed knowledge pool that organizes static and dynamic scene elements for synchronized reasoning, and (iii) a Retrieval-Augmented Generation (RAG) mechanism that grounds decisions in structured, real-time environmental knowledge. Together, these components directly address format fragmentation, temporal desynchronization, and weak grounding, providing a scalable bridge between collaborative sensing and knowledge-driven planning. Our contributions are threefold:

- We introduce **V2X-UniPool**, the first **knowledge-driven language-based V2X framework** for AD systems in scenario-aware, smarter decision making. The framework contains: (i) a language translation module, (ii) a time-indexed knowledge pool, and (iii) contextualized RAG grounding.

- To address heterogeneity and synchronization issues, we construct a **time-indexed knowledge pool** that integrates infrastructure-side sensor records into a unified, language-based representation. Accompanied by temporal static and dynamic data, the pool enables efficient storage, real-time retrieval, and temporally consistent reasoning.

- V2X-UniPool incorporates a **RAG mechanism** that allows models to retrieve this time-indexed knowledge pool for relevant scene context. This real-time grounding effectively bridges the modality gap and reduces hallucinations.

- Experiments on the real-world DAIR-V2X dataset demonstrate that V2X-UniPool achieves **state-of-the-art planning accuracy and safety**, while reducing communication cost by more than 80% and attaining the **lowest overhead**, even under zero-shot vehicle-side model.

## 2 RELATED WORK

### 2.1 V2X DATA FUSION IN COOPERATIVE PERCEPTION

Vehicle-independent perception, which relies solely on onboard sensors and intra-vehicle fusion, often suffers from limited field of view and occlusions, making it difficult to ensure reliable situation awareness in dense or unstructured environments (Miao et al., 2022). In contrast, cooperative perception—enabled by Vehicle-to-Everything (V2X) communication—facilitates the sharing of sensory information among vehicles and infrastructure, leading to a more comprehensive understanding of the traffic environment (Han et al., 2023; Yang et al., 2022; Liu et al., 2023). This approach has proven particularly valuable in high-occlusion scenarios, where external perspectives can compensate for blind spots and improve safety-critical decision making (Narri et al., 2021; Xiao et al., 2023; Loh et al., 2024).

Data fusion plays a vital role in this approach (Ren et al., 2022; Han et al., 2023). Its methods are commonly divided into three categories: early fusion, intermediate fusion, and late fusion. Early fusion aggregates raw sensor data across agents before any local processing, offering fine-grained cooperation but incurring high bandwidth and strict temporal synchronization requirements (Chen et al., 2019; Arnold et al., 2020). Intermediate fusion shares encoded features among agents to achieve a balance between communication cost and accuracy, and has therefore become a widely adopted paradigm (Liu et al., 2023; Wang et al., 2020; Xu et al., 2022; Wang et al., 2025). However, it faces major challenges in heterogeneous environments (Gao et al., 2025b; Huang et al., 2023). In contrast, late fusion—especially when implemented through LLM-based reasoning—can better accommodate agent heterogeneity by deferring integration to a high-level semantic space (Chiu et al., 2025; Gao et al., 2025a).

In this work, we treat shared outputs as language-level abstractions rather than transmitting raw data or intermediate features as in prior data-fusion methods. This design not only significantly reduces communication cost but also supports efficient, interpretable reasoning under realistic bandwidth constraints.

### 2.2 KNOWLEDGE-EMPOWERED AUTONOMOUS DRIVING

The rapid advancement of language models is reshaping the field of AD, enabling a shift from traditional, situation-aware pipelines to more holistic and knowledge-driven paradigms. By leveraging their ability to understand complex driving scenes, they are increasingly regarded as foundational components in modern AD architectures (Zhu et al., 2024; Luo et al., 2024). Their powerful contextual reasoning capabilities allow AD systems to interpret intricate traffic scenarios and support

more informed decision-making. Recent studies have demonstrated that language models not only enhance perception but also contribute to downstream tasks such as trajectory prediction and motion planning, thereby improving the overall safety, adaptability, and intelligence of AD systems (Yang et al., 2023; Guo et al., 2024).

Building on this foundation, recent work has shown the potential of language models to support end-to-end systems from initial sensing and scene understanding to planning and control (Cui et al., 2024). However, their reasoning capacity is often constrained by static pre-training, which restricts real-time adaptability and environmental grounding (Li et al., 2023). To address this, the RAG paradigm has emerged as a promising enhancement, allowing models to proactively query external databases for up-to-date and context-specific knowledge (Fan et al., 2024). Systems such as RAG-Driver (Yuan et al., 2024) and RAG-Guided (Yu et al., 2024) demonstrate how integrating RAG can reduce hallucinations, improve decision consistency, and expand the effective knowledge scope of LLMs in autonomous driving contexts (Luo et al., 2025).

In contrast to prior knowledge-driven AD that suffers from hallucinations and lacks environmental grounding, our framework anchors reasoning in a structured knowledge pool, providing temporally aligned, interpretable, and query-efficient context. Furthermore, V2X-UniPool enables precise and proactive retrieval of spatiotemporal knowledge tailored to the ego vehicle's state, ensuring both consistency and real-time responsiveness.

## 3 METHODOLOGY

### 3.1 FRAMEWORK OVERVIEW

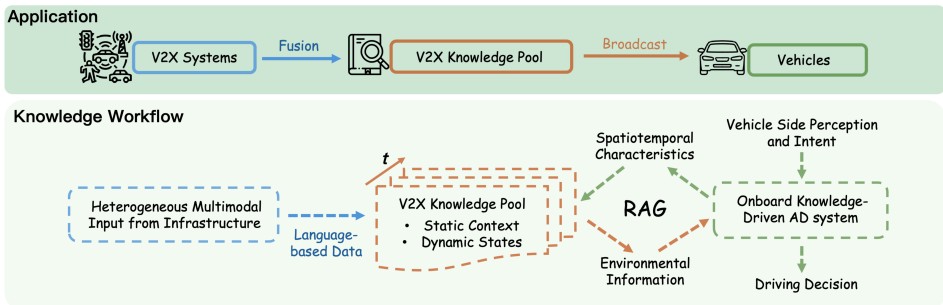

Figure 1: Framework of V2X-UniPool

To achieve extended situational awareness and accurate reasoning, we introduce V2X-UniPool, a unified multimodal framework deployed on the infrastructure side. This framework bridges the gap between raw multimodal sensor data from real-world infrastructure and structured, language-based representations. It continuously aggregates heterogeneous multimodal inputs into a temporally organized V2X Knowledge Pool, while the onboard knowledge-driven AD system interacts with this pool through a RAG mechanism. At each reasoning step, the infrastructure encodes the latest knowledge into a compact language-based snapshot, which is broadcast; the vehicle then obtains environmental information conditioned on its spatiotemporal characteristics. This fused representation supports anticipatory planning, thereby enabling real-time, grounded, and adaptive reasoning in complex traffic scenarios.

The core of this framework is a time-indexed **V2X Knowledge Pool**, which serves as a comprehensive repository for environmental scene understanding. Environmental scene understanding refers to the perception system's ability to extract, recognize, and interpret features of all surrounding elements relevant to driving (Muhammad et al., 2022). In this paper, the V2X Knowledge Pool formalizes such scene understanding as an integration of language-based representations of the surrounding world, encompassing both static elements (e.g., road geometry, traffic signs, map features) and dynamic elements (e.g., vehicles, traffic lights). Accordingly, the V2X Knowledge Pool is organized into a Static Pool and a Dynamic Pool.

**RAG reasoning mechanism** is integrated with the V2X Knowledge Pool. At each iteration, the pool broadcasts a language-based snapshot from this pool, and the ego vehicle retrieves relevant environ-

mental information based on timestamp and location. This retrieved knowledge that covers static context and dynamically updated traffic states is then fused into a unified representation together with the local sensory input of the vehicle. The fused context and local perception are subsequently processed by the vehicle-side model to generate the final driving decisions.

## 3.2 V2X Knowledge Pool Construction

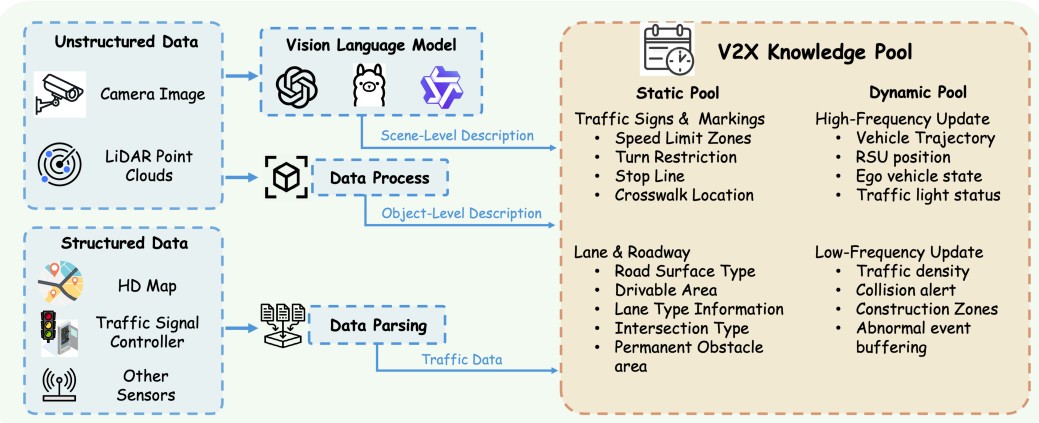

Figure 2: Overview of the V2X Knowledge Pool Construction

**V2X Raw Data Processing** To construct a temporally aligned and semantically structured knowledge representation, the raw sensor data are first transformed into language-based data. These raw data are categorized into two modalities: **Unstructured Data**, including camera images and LiDAR point clouds; and **Structured Data**, comprising HD maps, traffic signal phases, average flow speed, vehicle counts, density metrics, and other infrastructure sensors. This focus leverages the global field-of-view and fixed positioning of infrastructure sensors, offering more comprehensive and stable environmental coverage than the ego vehicle's limited and dynamic perception. All collected raw data are temporally synchronized and spatially calibrated within a unified coordinate frame to ensure cross-modality alignment. The processing pipeline is divided according to data modality and preprocessing requirements:

For Unstructured Data, camera and LiDAR inputs from roadside units are converted into interpretable semantic representations. High-resolution bird's-eye-view (BEV) images are denoised, standardized, and normalized before being processed by a state-of-the-art vision-language model (GPT-4o 2024). Task-specific prompts guide the model to extract scene-level semantics, outputting structured descriptions that include a `reason` field (natural-language explanation) for interpretability and a `prediction` field to infer short-term state transitions. This reasoning-aware design improves both transparency and utility (Wei et al., 2022; Rajani et al., 2019; Bai et al., 2024). In parallel, LiDAR point clouds are geometrically filtered to remove noise and normalized for consistent density and frame alignment. Combined with image semantics, these 3D spatial cues enhance the detection and localization of road users and infrastructure elements, yielding a consistent representation of the traffic scene. For Structured Data, inputs are cleaned, temporally synchronized, and spatially aligned. Relevant fields are parsed, missing values are interpolated or replaced with defaults chosen based on infrastructure priors or domain-specific heuristics, and numerical fields are standardized. These records are then matched with unstructured data by timestamp and location, enabling unified integration.

The resulting dataset comprises temporally aligned, semantically normalized heterogeneous data, where both structured and unstructured modalities are transformed into a unified language-based representation. This forms the basis of the V2X Knowledge Pool, supporting interpretable and efficient reasoning across complex traffic scenarios.

**Static Pool** The Static Pool of the V2X Knowledge Pool is defined as the collection of infrastructure-level environmental semantics whose temporal dynamics are negligible within operational planning horizons (Huang et al., 2023). These elements are effectively time-invariant in practice, as their variations remain below a stability threshold $\epsilon$ during short time intervals, thereby

providing a reliable basis for reasoning and decision-making across time-aligned traffic scenarios. Formally, a data element $\mathcal{D}_{\text{static}}$ is included in the Static Pool if its feature representation satisfies Eq 1:

$$\left| \frac{\partial f(\mathcal{D}_{\text{static}})}{\partial t} \right| < \epsilon, \quad \forall t \in [t_0, t_0 + T] \tag{1}$$

where $f(\cdot)$ denotes the element's numerical representation, $\epsilon \ll 1$ is a predefined stability threshold, and $T$ is the planning horizon.

The Static Pool is typically constructed offline or during system initialization for a given region and updated only when long-term infrastructure changes occur (e.g., lane reconfiguration, construction updates, or map revisions). Each entry is represented in a language-based format, enabling interpretable and semantically grounded descriptions of the environment, including traffic signs, lane markings, and roadway geometry.

By maintaining a stable, high-level abstraction of the physical scene, the Static Pool functions as the long-term memory of the reasoning system. It ensures consistent grounding for dynamic perception, supports scenario understanding, and reduces the computational burden of repeatedly processing static semantics in real time.

**Dynamic Pool** The Dynamic Pool of the V2X Knowledge Pool captures temporally evolving traffic states. Unlike the Static Pool that includes invariant infrastructure, the Dynamic Pool is continuously updated to reflect changes in traffic participants, signal states, and environmental conditions. Formally, a data element $\mathcal{D}_{\text{dynamic}}$ is classified as part of the Dynamic Pool if it violates the constraint of temporal invariance and exhibits perceptible change within the planning horizon as Eq 2:

$$\left| \frac{\partial f(\mathcal{D}_{\text{dynamic}})}{\partial t} \right| \geq \epsilon, \quad \exists t \in [t_0, t_0 + T] \tag{2}$$

where $f(\cdot)$ denotes the numerical feature representation of the element, and $\epsilon$ is the minimum change rate required to be considered temporally dynamic. The threshold is task-defined and reflects the system's sensitivity to environmental variation. This definition ensures that only non-stationary elements requiring timely updates are included in the Dynamic Pool, enabling responsive and adaptive planning.

The Dynamic Pool is constructed online and continuously updated to capture temporally evolving states, and to balance responsiveness and semantic granularity, we divide it into two sub-pools: a **High-Frequency Update Sub-Pool** and a **Low-Frequency Update Sub-Pool**.

**High-Frequency Update Sub-Pool** contains data streams that change rapidly over short time intervals and are critical for short-term prediction and motion planning. This includes object trajectories, velocities, types, and traffic light states updated at rates of at least 10 Hz. Formally, a dynamic data instance $\mathcal{D}_{\text{dynamic}}$ is assigned to the High-Frequency Update Sub-Pool if it satisfies Eq 3:

$$\left| \frac{\partial f(\mathcal{D}_{\text{dynamic}})}{\partial t} \right| \geq \epsilon_{\text{high}}, \quad f_s(\mathcal{D}_{\text{dynamic}}) \geq 10\,\text{Hz} \tag{3}$$

where $\epsilon_{\text{high}}$ denotes the minimum rate of change for highly dynamic content, and $f_s(\cdot)$ represents the sampling frequency. This condition ensures that only fast-evolving, high-resolution data are routed to this sub-pool for real-time planning.

**Low-Frequency Update Sub-Pool** stores moderately dynamic yet semantically rich information that evolves over slower time scales. This includes aggregated traffic density, collision alerts, construction updates, and abnormal events, typically updated at rates around 1 Hz. Formally, a dynamic data instance $\mathcal{D}_{\text{dynamic}}$ is assigned to the Low-Frequency Update Sub-Pool if it satisfies Eq 4:

$$\epsilon_{\text{low}} \leq \left| \frac{\partial f(\mathcal{D}_{\text{dynamic}})}{\partial t} \right| < \epsilon_{\text{high}}, \quad f_s(\mathcal{D}_{\text{dynamic}}) < 10\,\text{Hz} \tag{4}$$

where $\epsilon_{\text{low}}$ and $\epsilon_{\text{high}}$ define the acceptable range of perceptual change for moderate dynamics. Elements with variation below $\epsilon_{\text{low}}$ are instead assigned to the Static Pool, ensuring a clear separation between static and dynamic semantics. This guarantees that mid-rate signals are preserved for semantic fusion and planning over longer horizons.

Together, these two sub-pools provide a comprehensive multi-resolution temporal representation of the current traffic environment, enabling knowledge-driven AD to anticipate, reason, and respond adaptively under dynamic conditions.

## 3.3 RAG-BASED REASONING

We design a RAG-based collaborative reasoning mechanism that enables the ego vehicle to continuously access infrastructure-side knowledge while preserving temporal and spatial relevance.

**Language-based Knowledge Broadcasting** The infrastructure encodes the aggregated contents of the Static Pool, the High-Frequency Update Sub-Pool, and the Low-Frequency Update Sub-Pool into a compact language-based snapshot $S_t$ as Eq 5, which is then broadcast to nearby vehicles rather than transmitting the entire pool history:

$$S_t = \text{Encode}\Big( P^{\text{Static}} \oplus P^{\text{HF}} \oplus P^{\text{LF}} \Big), \tag{5}$$

where $P^{\text{Static}}$, $P^{\text{HF}}$, and $P^{\text{LF}}$ denote concatenation-based aggregation from the respective sub-pools, and $\oplus$ represents modality-wise concatenation.

**Environmental Information Retrieval** Upon receiving the broadcast stream $S_t$, the ego vehicle filters the knowledge based on its spatiotemporal state. Specifically, it further extracts the information most relevant to its current timestamp $t$ and location $l$ as Eq 6:

$$E_{t,l} = \text{Retrieve}\big(t, l;\ S_t\big), \tag{6}$$

where $\text{Retrieve}(\cdot)$ returns spatiotemporally filtered knowledge such as nearby objects, traffic lights, road conditions, aggregated traffic density, and abnormal events. This design ensures temporal consistency while filtering the knowledge to what is most relevant for the vehicle.

**Joint Perception and Reasoning** The retrieved environmental information $E_{t,l}$ is then combined with the vehicle's local sensory input $V$. In practice, the fusion is implemented through structured prompt construction. The vehicle-side language model then performs reasoning over this fused representation to generate the final driving decision $\hat{D}$ as Eq 7:

$$\hat{D} = \text{LM}(\text{Fuse}(V, E_{t,l})). \tag{7}$$

This RAG-based design ensures that each vehicle always reasons with the most spatiotemporally relevant infrastructure knowledge, while the language-based snapshot representation minimizes communication overhead and provides interpretable semantics for complex urban scenarios.

## 4 EXPERIMENT

### 4.1 EXPERIMENT SETTINGS

We conduct comprehensive evaluations of V2X-UniPool on the DAIR-V2X-Seq dataset (Yu et al., 2023), a large-scale real-world V2X benchmark featuring both vehicle-side and infrastructure-side sensors. The dataset comprises sequential perception and trajectory forecasting subsets. The sequential perception dataset includes over 15,000 frames across 95 scenarios, collected at 10 Hz from vehicle and roadside sensors. The trajectory forecasting dataset is significantly larger, with 210,000 scenarios from 28 intersections, including 50,000 cooperative-view, 80,000 ego-view, and 80,000 infrastructure-view cases. Each scenario is supplemented with 10-second trajectories, high-definition vector maps, and real-time traffic light signals. This comprehensive setup enables fine-grained exploration of cooperative perception and planning, allowing assessment of infrastructure-enhanced reasoning in diverse urban traffic conditions.

All models follow the OpenEMMA-style planning formulation (Xing et al., 2025), where each input consists of front-view images and a history of ego vehicle speeds and curvatures, and the output is a prediction of future speed and curvature vectors. This design reflects real-world driving requirements and supports precise evaluation of planning performance. For the experiment, we construct over 10,000 ego-view scenarios from DAIR-V2X-Seq, each spanning 10 seconds and formatted as structured planning prompts with historical motion states and aligned future trajectory labels. During inference, all vehicle-side models receive the same V2X-UniPool RAG retrieval results, ensuring that observed performance differences are solely attributed to the models' reasoning ability over external semantic context.

Beyond the dataset setup and input–output design, our experiments first benchmark V2X-UniPool against representative planning baselines, jointly evaluating planning accuracy, safety, and communication efficiency together with a detailed latency analysis of the infrastructure-to-vehicle pipeline.

We then assess generality through ablations across diverse vehicle-side models and examine scaling behavior within the Qwen-2.5-VL family. Together, these results provide a comprehensive evaluation of accuracy, safety, efficiency, and scalability for V2X-UniPool.

## 4.2 EXPERIMENT RESULTS

Table 1: Planning Evaluation Results. V2X-UniPool integrates external knowledge via structured RAG reasoning. The vehicle-side model is based on Qwen-2.5-vl-7B.

| Method | L2 Error (m) ↓ | | | | Collision Rate (%) ↓ | | | | Transmission Cost ↓ |
|---|---|---|---|---|---|---|---|---|---|
| | 2.5s | 3.5s | 4.5s | Avg. | 2.5s | 3.5s | 4.5s | Avg. | |
| V2VNet (Wang et al., 2020) | 2.31 | 3.29 | 4.31 | 3.30 | 0.00 | 1.03 | 1.47 | 0.83 | $8.19 \times 10^7$ |
| CooperNaut (Cui et al., 2022) | 3.83 | 5.26 | 6.69 | 5.26 | 0.59 | 1.92 | 1.63 | 1.38 | $8.19 \times 10^7$ |
| UniV2X - Vanilla (Yu et al., 2025) | 2.21 | 3.31 | 4.46 | 3.33 | 0.15 | 0.89 | 2.67 | 1.24 | $8.19 \times 10^7$ |
| UniV2X (Yu et al., 2025) | 2.60 | 3.44 | 4.36 | 3.00 | 0.00 | 0.74 | 0.49 | 0.41 | $8.09 \times 10^5$ |
| V2X-VLM (You et al., 2024) | 1.09 | 1.12 | 1.42 | 1.21 | 0.02 | 0.03 | 0.03 | 0.03 | $1.24 \times 10^7$ |
| **V2X-UniPool** | **0.95** | 1.44 | 2.02 | 1.47 | **0.00** | **0.02** | 0.05 | 0.04 | $\mathbf{1.51 \times 10^5}$ |

Table 1 reports the overall planning and communication results. **V2X-UniPool** achieves the best short-horizon planning accuracy (**0.95 m at 2.5s**) and sustains competitive average error (1.47 m), comparable to the state-of-the-art V2X-VLM (1.21 m). In terms of safety, it consistently maintains a near-zero collision rate (**0.00%** at 2.5s, **0.02%** at 3.5s, average 0.04%), matching the reliability of V2X-VLM (0.03%). These results highlight that V2X-UniPool is capable of delivering both accurate and safe planning, even under challenging horizons.

Beyond accuracy and safety, V2X-UniPool establishes a new benchmark in efficiency. It reduces transmission cost to only $\mathbf{1.51 \times 10^5}$ bytes, which is over two orders of magnitude smaller than V2X-VLM ($1.24 \times 10^7$) and three orders lower than traditional cooperative perception approaches ($8.19 \times 10^7$). This drastic reduction makes V2X-UniPool significantly more deployable under realistic bandwidth constraints.

To further quantify the practical efficiency of this design, we conduct a detailed latency breakdown analysis. This evaluation decomposes the infrastructure-to-vehicle pipeline into its constituent stages, allowing us to measure the contribution of raw data preprocessing, temporal indexing, communication, and vehicle-side retrieval. These results demonstrate that by transmitting compact language-based knowledge snapshots, V2X-UniPool not only achieves the lowest communication cost, but also realizes a lightweight end-to-end pipeline, incurring negligible overhead beyond data preprocessing.

Table 2: Latency breakdown of V2X-UniPool. The total latency corresponds to the infrastructure-to-vehicle processing time for one pipeline, including communication (estimated with 100 Mbps link).

| Process | Description | Latency (ms) | Proportion (%) |
|---|---|---|---|
| Preprocessing | V2X Raw Data Processing (parallel, max sensor) | ~250 | 82.8 |
| Knowledge Pool Ops | Temporal indexing + pool update | ~40 | 13.2 |
| Communication (100 Mbps) | Transmission delay per broadcast | ~12 | 4.0 |
| Vehicle Extraction | Query relevant pool entries (retrieval) | ≈0.0 | <0.1 |
| Prompt Construction | Build structured input for vehicle-side model | ≈0.0 | <0.1 |
| **Total** | - | **~302** | **100.0** |

Table 2 reports the latency breakdown of V2X-UniPool. The total processing takes approximately **302 ms**, where the dominant component is raw V2X data preprocessing (**250 ms**, **82.8%**). Notably, this latency reflects the maximum among parallel sensor streams, since preprocessing for different modalities is executed concurrently. Knowledge pool operations, including temporal indexing and updates, add only **40 ms** (**13.2%**), while vehicle-side retrieval and prompt construction incur negligible overhead (<0.1%). In terms of communication, with 5G NR-V2X links (typical throughput approaching 100 Mbps), V2X-UniPool requires only **12 ms** per broadcast (**4.0%**). On the vehicle side, the latency for reasoning depends on the deployed model size. Nonetheless, V2X-UniPool

consistently ensures that the *infrastructure-to-vehicle* operations remain lightweight, leaving the computation–latency trade-off mainly determined by the chosen vehicle-side model.

### 4.3 ABLATION STUDY

To systematically evaluate the effectiveness of the proposed V2X-UniPool, two complementary ablation studies are designed with the goal of assessing its benefits across both *different model architectures* and *different model scales*. (i) *Cross-model evaluation*: We compare the planning performance of representative vehicle-side language models with and without V2X-UniPool integration. The evaluation spans both proprietary and open-source paradigms, including **GPT-4o (2024)**, **GPT-4.1 Mini**, **Gemini-2.0**, and **Llama-3.2**. (ii) *Scaling behavior*: To examine how V2X-UniPool interacts with model capacity, we further conduct experiments across multiple scales of the same architecture. Specifically, we adopt the **Qwen-2.5-VL** family at four parameter sizes (**3B**, **8B**, **32B**, and **72B**) to analyze scaling trends under both baseline and V2X-UniPool enhanced settings. This two-fold design enables us to disentangle the effects of model scaling from those of structured external knowledge, and to demonstrate the generality and scalability of V2X-UniPool.

Table 3: Comprehensive ablation study on different vehicle-side models with and without V2X-UniPool. Metrics include **L2 Error** (m, lower is better), **Collision Rate** (%, lower is better), and **Comfort Score** (higher is better), reported at 2.5s, 3.5s, and 4.5s horizons.

| Model | Metric | w/o V2X-UniPool | | | w/ V2X-UniPool | | |
|---|---|---|---|---|---|---|---|
| | | 2.5s | 3.5s | 4.5s | 2.5s | 3.5s | 4.5s |
| GPT-4o (2024) | Error ↓ | 1.25 | 1.93 | 2.74 | **1.18** | **1.81** | **2.56** |
| | Collision ↓ | 0.00 | 0.00 | 0.05 | 0.00 | 0.00 | **0.04** |
| | Comfort ↑ | 0.52 | 0.54 | 0.56 | **0.63** | **0.65** | **0.66** |
| GPT-4.1 Mini | Error ↓ | 1.45 | 2.21 | 3.11 | **1.41** | **2.18** | **3.10** |
| | Collision ↓ | 0.00 | 0.00 | 0.10 | 0.00 | 0.00 | **0.01** |
| | Comfort ↑ | 0.58 | 0.61 | 0.62 | **0.59** | 0.61 | **0.63** |
| Gemini-2.0 | Error ↓ | 1.37 | 2.18 | 3.17 | **1.29** | **2.01** | **2.92** |
| | Collision ↓ | 0.00 | 0.05 | 0.05 | 0.00 | **0.04** | **0.04** |
| | Comfort ↑ | 0.30 | 0.33 | 0.34 | **0.33** | **0.36** | **0.37** |
| Llama-3.2 | Error ↓ | 1.26 | 1.92 | 2.72 | **1.24** | **1.90** | **2.69** |
| | Collision ↓ | 0.07 | 0.07 | 0.11 | **0.02** | **0.03** | **0.03** |
| | Comfort ↑ | 0.53 | 0.56 | 0.58 | 0.51 | 0.56 | **0.59** |

While collision rate is a critical safety indicator, for most strong models it already remains near zero, leaving limited room for further reduction. Therefore, we additionally introduce a **Comfort Score** to better capture improvements in trajectory smoothness and ride quality. Beyond safety, passenger experience is a critical factor for autonomous vehicles, as the primary stakeholders are the passengers themselves. To capture the smoothness of the trajectory and the comfort of the ride, we adopt a differentiable metric based on longitudinal acceleration and jerk, two key factors widely recognized in studies of transportation safety and comfort (ISO, 1997; Bae et al., 2020; Yan et al., 2021). Here, $\ddot{v}$ denotes longitudinal acceleration and $\dddot{v}$ is jerk (the rate of change of acceleration). The averages are computed over $n$ predicted future frames. Following prior literature, we set $\alpha = 1.0$ and $\beta = 1.0$ to balance the contributions of acceleration and jerk. The resulting score lies within $[0, 1]$, where higher values indicate smoother and more comfortable planned trajectories. The score function is defined as Eq 8:

$$\text{ComfortScore} = 1 - \tanh\left(\alpha \cdot |\bar{\ddot{v}}| + \beta \cdot |\bar{\dddot{v}}|\right) \tag{8}$$

From Table 3, it is clear that **V2X-UniPool consistently enhances all evaluated models across the three key metrics**. On average, the *L2 Error* decreases by about **1–8%** across horizons (mean: 4–5%), confirming more accurate trajectory generation. The *Comfort Score* improves steadily for most models, with average gains of **0.035**, reflecting smoother driving behavior. The most pronounced effect, however, is observed in the *Collision Rate*: while GPT-4o and Gemini-2.0 achieve modest reductions, Llama-3.2 shows the most striking improvement, with its collision rate dropping from 0.11 to 0.03 at the 4.5s horizon equivalent to a **73% reduction**. This indicates that V2X-UniPool not only improves planning accuracy, but also provides substantial safety benefits. Overall,

the ablation results highlight that the integration of V2X-UniPool leads to more precise trajectories, significantly lower collision risks, and improved comfort, with safety improvement being the most prominent gain.

We evaluate the effect of V2X-UniPool on model scaling using the Overall Average ADE across same 100 driving scenarios. As shown in Table 4, incorporating V2X-UniPool consistently reduces planning error across all model sizes. For example, Qwen-3B with UniPool improves from 1.65 to 1.59, effectively narrowing the gap to the 7B base model (1.37). Similarly, Qwen-32B with UniPool achieves 1.27,

Table 4: Overall Average ADE comparison of Qwen models with and without V2X-UniPool.

| Model | Base | +V2X-UniPool |
|---|---|---|
| Qwen-3B | 1.6482 | 1.5851 |
| Qwen-7B | 1.3699 | 1.3456 |
| Qwen-32B | 1.3803 | 1.2671 |
| Qwen-72B | 1.3241 | 1.2998 |

surpassing the performance of the 72B base model (1.32). These results highlight that V2X-UniPool enables smaller models to approach or even exceed the performance of much larger counterparts, thereby achieving near state-of-the-art accuracy at substantially lower computational cost.

Building on these error-based comparisons, we further examine how improvements propagate into downstream driving quality metrics, including comfort and collision. Figure 3 summarizes these effects across the Qwen-2.5-VL family. Overall, **V2X-UniPool brings consistent benefits across all model scales**, with the most notable gains achieved by **Qwen-3B** and **Qwen-32B**. Among the evaluation metrics, *Error* and *Comfort* show the most substantial improvements, indicating that external structured knowledge not only enhances prediction accuracy but also contributes to smoother and more stable driving behavior, while collision reductions are comparatively smaller due to the already low baseline rates.

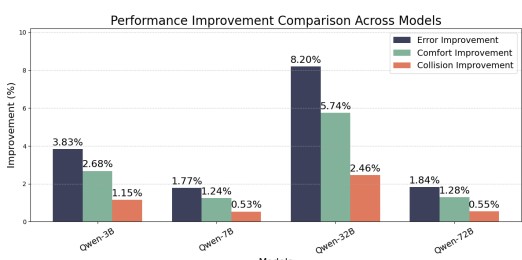

Figure 3: Relative improvements with V2X-UniPool across the Qwen-2.5-VL family. **Left:** Error reduction (%). **Middle:** Comfort improvement (%). **Right:** Collision reduction (%).

## 5 CONCLUSION

This study presents **V2X-UniPool**, a unified framework for knowledge-driven AD system. By translating raw multimodal V2X signals into structured, language-based knowledge and organizing them in a temporally indexed pool, V2X-UniPool enables contextualized, RAG-based reasoning on the vehicle side. Experiments on the real-world DAIR-V2X dataset show that it improves planning accuracy and safety while reducing communication cost by more than 80%, establishing a new paradigm that integrates V2X collaboration with language-model reasoning. The results further highlight its practical potential, achieving state-of-the-art accuracy and safety with the lowest communication overhead and small end-to-end latency, while consistently benefiting models of different architectures and scales to ensure strong scalability.

**Limitations and broader impacts.** V2X-UniPool is designed as an augmentation layer: it consistently improves planning quality, but its complete latency and final performance inevitably depend on the capacity of the underlying AD model. While our current evaluation is based on open-loop experiments, the next step is to advance toward closed-loop validation. Encouraged by the positive feedback from current results, we are in the process of deploying V2X-UniPool in a closed-loop setting to further assess its robustness under interactive traffic dynamics. We expect this study to advance the integration of V2X systems with knowledge-driven AD, paving the way for safer, more cost-effective, and highly scalable driving applications.

ETHICS STATEMENT

This research is based on publicly available datasets (DAIR-V2X). No personally identifiable or sensitive data were used. All experiments are conducted in a simulation or dataset-based environment, ensuring no direct interaction with real-world human subjects or safety-critical systems. The study aims to advance autonomous driving safety and efficiency without compromising privacy or ethical standards.

REPRODUCIBILITY STATEMENT

We ensure reproducibility by providing detailed methodology, dataset descriptions, and evaluation protocols in the paper. An anonymous GitHub repository is made available at https://anonymous.4open.science/r/V2X-UniPool-7326, containing code, configuration files, and experiment scripts for replication.

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

# A APPENDIX

## .1 AUTHOR STATEMENT ON LLM USAGE

In accordance with ICLR policies on the use of large language models (LLMs), we clarify the scope of LLM assistance in this work. LLMs were used for language refinement and editing purposes, such as improving grammar, style, and readability of the manuscript text. All research ideas, framework design, experimental setup, implementation, and analysis were conceived and conducted independently by the authors without reliance on LLMs for technical content generation. The use of LLMs did not influence the methodology, experimental results, or scientific conclusions. Their role was strictly limited to polishing the presentation of the paper.

## .2 VISION-LANGUAGE MODEL PROMPTS AND EVALUATION ON UNSTRUCTURED DATA: IMAGES

In this section, we present the system prompts designed for traffic scene understanding and the corresponding evaluation results on unstructured image data. The goal is to assess how well vision-language models (VLMs) can translate visual content into structured semantic representations that serve as a reliable foundation for downstream reasoning and decision-making.

### .2.1 SYSTEM PROMPT DESIGN

The following system prompt was used to guide the VLM for structured traffic scene analysis:

Listing 1: VLM System Prompt Schema

```
You are a vision-language assistant for traffic scene understanding.
Given an image, analyze and extract structured information in the
following JSON format.

Each field should be accurate, concise, and use just a few words or a
short phrase. Include sample-style content. All location descriptions
use the traffic light as the absolute reference point. Here's the schema
and example values:

{
  "lane_markings": {
    "crossing": [ {"location": "across from traffic light",
               "function": "pedestrian crossing",
               "confidence": 0.95} ],
    "straight_lane": [ {"location": "directly in front of traffic light",
                    "function": "go straight",
                    "confidence": 0.95} ],
    "right_turn_lane": [ {"location": "to the right of traffic light",
                      "function": "right turn",
                      "confidence": 0.95} ],
    "left_turn_lane": [ {"location": "to the left of traffic light",
                     "function": "left turn",
                     "confidence": 0.95} ]
  },

  "traffic_sign": [ {"location": "adjacent to traffic light on the right",
                "sign_type": "speed limit",
                "text_info": "Speed Limit 40"} ],

  "road_surface_condition": {
    "surface_type": "asphalt",
    "condition": "dry",
    "surface": "dip"
  },

  "daylight_condition": {"lighting": "daylight"},
  "weather_condition": {"weather_type": "clear", "intensity": "none"},
```

```
  "traffic_flow": {"congestion_level": "low"},

  "vulnerable_road_users": [ {"type": "child",
                       "location": "near crosswalk by traffic light",
                       "action": "waiting",
                       "visibility": "partial",
                       "confidence": 0.87} ],

  "collision_alert": [ {"location": "in right lane just past traffic
      light",
                    "involved_objects": "car and cyclist",
                    "severity": "high",
                    "status": "predicted"} ],

  "emergency_condition": [ {"type": "ambulance",
                       "location": "in left lane approaching traffic
                          light",
                       "siren": "on",
                       "priority_action": "yield required"} ],

  "construction_condition": [ {"location": "right lane near traffic
      light",
                          "type": "road work",
                          "warning_sign": "construction ahead",
                          "closed_lane": "right lane"} ]
}
```

## .2.2 EXPERIMENTAL RESULTS: FIELD-WISE ACCURACY

To evaluate the quality of scene-level semantic extraction from multimodal inputs, we conducted multiple rounds of testing using several state-of-the-art language models. Specifically, each extracted semantic field from the image inputs was automatically reviewed and scored by VLMs, including GPT-4o and DeepSeek-VL r1, based on consistency with the visual evidence. Scores were assigned on a normalized 0–1 scale.

To ensure robustness, we averaged results across multiple runs and multiple evaluators. The final accuracy values reflect both cross-model agreement and field-level semantic fidelity. The aggregated scores for each semantic category are summarized in Table 5.

Table 5: Field-wise semantic extraction accuracy of VLMs under the designed prompt.

| No. | Field Name | Accuracy (0–1) |
|---|---|---|
| 1 | lane_markings | 0.996 |
| 2 | traffic_sign | 0.648 |
| 3 | road_surface_condition | 0.998 |
| 4 | daylight_condition | 1.000 |
| 5 | weather_condition | 0.993 |
| 6 | traffic_flow | 0.936 |
| 7 | vulnerable_road_users | 0.833 |
| 8 | collision_alert | 0.805 |
| 9 | emergency_condition | 0.999 |
| 10 | construction_condition | 0.940 |
| **Average** | **Overall Accuracy** | **0.915** |

This detailed field-wise analysis provides strong empirical evidence that the semantic layer accurately captures the intended scene understanding, forming a reliable basis for downstream reasoning and decision-making.

