# OpenReview forum: "V2X-UniPool: Unifying Multimodal Perception and Knowledge Reasoning for Autonomous Driving"
_ICLR.cc/2026/Conference — Submitted to ICLR 2026_

### Official Review · Reviewer_3myK · 2025-10-26

**Soundness:** 3
**Presentation:** 3
**Contribution:** 3
**Rating:** 6
**Confidence:** 3

**Summary:**

V2X-UniPool unifies V2X perception with language-based reasoning by translating multimodal signals into a time-indexed knowledge pool and using RAG to fuse the retrieved context with on-board perception for planning. This results in impressive improvements on DAIR-V2X, cutting transmissions by >80% and reducing collisions to near-zero.

**Strengths:**

* V2X-UniPool bridges perception-centric V2X with language-centric planning via a time-indexed knowledge pool and RAG grounding, explicitly tackling heterogeneity, temporal desynchronization, and hallucination.
* The static/dynamic pool provides multi-resolution temporal semantics, and ablations across model architectures show consistent gains.

**Weaknesses:**

* The latency and network assumptions may be optimistic. There’s no robustness study for lower bandwidth, variable latency, or packet loss.
* Results are reported only on DAIR-V2X-Seq and in open-loop. the paper note end performance still depends on the chosen vehicle-side AD model and plan to pursue closed-loop validation, so real-world robustness and controller interaction effects remain untested.

**Questions:**

The more complete illustration based on the latency and network assumptions will be helpful.

---

> ### Author Response · Authors · 2025-11-17
>
> We sincerely thank the reviewer for the thoughtful and constructive feedback. We greatly appreciate the recognition of **V2X-UniPool**’s contributions in bridging perception-centric and language-centric V2X reasoning. The reviewer’s comments on *closed-loop validation* and *network assumptions* are highly valuable and well-taken.
>
> (1) **Closed-loop validation.**
> We agree that validating V2X-UniPool in a closed-loop setup is an important next step. Our current study focuses on open-loop perception–reasoning evaluation to ensure reproducibility and isolate semantic contributions. A closed-loop integration with vehicle controllers and simulators (CARLA) is already in progress and will be included in our future work to assess real-world robustness and controller interactions.
>
> (2) **Illustration based on latency and network assumptions.**
> We appreciate the reviewer’s request for a more complete illustration. In our framework, the network assumptions (bandwidth, latency, and packet transmission rate) primarily affect the *knowledge exchange pipeline* between vehicles and infrastructure. As summarized in Table 2, the total latency (≈302 ms) mainly arises from infrastructure-side preprocessing (≈250 ms, 82.8 %) and temporal knowledge pool operations (≈40 ms, 13.2 %), while transmission latency under a 100 Mbps link contributes less than 5 %. Communication occurs in two stages: (i) infrastructure-side sensor data transmission to the knowledge pool, and (ii) broadcasting structured knowledge from the infrastructure to vehicles.  We would like to emphasize that **latency modeling and communication protocol design are not the core contributions of this work**. These modules are included to demonstrate the **feasibility and deployability** of V2X-UniPool under realistic network conditions rather than to propose new networking mechanisms. To further clarify these assumptions, we will provide an additional figure in the supplementary material illustrating the **communication topology and latency propagation pipeline** of V2X-UniPool, including how network delays and bandwidth constraints influence the asynchronous knowledge update process. We believe this addition will make our system assumptions clearer and more reproducible.
>
> We thank the reviewer again for the constructive comments, which have helped us further clarify the system’s scope, limitations, and future directions.

---

### Official Review · Reviewer_2oTn · 2025-10-31

**Soundness:** 2
**Presentation:** 2
**Contribution:** 2
**Rating:** 2
**Confidence:** 5

**Summary:**

This paper introduces V2X-UniPool, a unified framework that bridges V2X perception with language-based reasoning for planning in autonomous driving. The system converts raw sensor streams from infrastructure and other various information into structured, language-based knowledge and organizes this knowledge into a knowledge pool with multiple categories. The empirical evaluation on the DAIR-V2X dataset demonstrates improvements in planning accuracy, safety, and communication cost over previous baselines.

**Strengths:**

1. The proposed framework, V2X-UniPool, using natural language as the only form to transmit information from the infrastructure to vehicles, significantly reduces the communication cost.
2. The design of Static/Dynamic, High Frequency/Low Frequency pool systematically organizes the information stored in the infrastructure.

**Weaknesses:**

### Major Weaknesses
1. **The performance gain compared with the previous SoTA (V2X-VLM) is not significant.**
- Compared with full configuration of V2X-VLM, the L2 error is much higher. Notably, while V2X intuitively should provide more performance gain in long-term planning (Just as the authors stated, V2X helps extend perception range), V2X-UniPool shows an even larger gap on 3.5s and 4.5s error.
- Compared with the no-fusion result from V2X-VLM, which achieves an average L2 of 1.49m, the 1.47m result is not convincing, especially when more external knowledge is given.
- Considered V2X-VLM directly predicts waypoints, V2X-UniPool predicts speed and curvature following the OpenEmma paradigm. The effect of this difference is unclear.
2. **Missing details in the knowledge pool component.**
- In section 1, section 3.2, and Figure 2, LiDAR input is seen as an essential part, but no further explanation on how this modality is integrated into the system.
- In section 3.2, from equations 3 and 4, it seems that the proposed two sub-pools cannot fully cover the dynamic pool described in Eq. 2.
3. **The ablation study is also not well designed to prove the effectiveness.**
- Multiple key components, including information retrieval, knowledge pool operations, build up a relatively complicated system. However, no detailed information is provided to show how those components contribute to the performance gain in Table 3.

### Minor Weaknesses
1. The VLM setup (probabilistic sampling or not, preset temperature) is unclear, which is important to the significance of the results in Table 3.
2. In section 3.3, Encode and Retrieve are not well defined.

**Questions:**

1. What's the performance of V2X-UniPool if the downstreamed AD model predicts waypoints directly? This helps to have a fair comparison with baselines.
2. Do you have qualitative results to show the actual improvement with V2X information integrated? Given that DAIR-V2X is collected at signalized intersections, do frequent conflicting interactions occur?
3. For the mapping from messages to specific pools, is it fixed during the design stage or dynamically changes on the fly? Will different pools be treated differently in the downstream tasks? How will this categorization affect the performance?

---

> ### Author Response · Authors · 2025-11-14
> **Reply about Clarification on System Design and Deployability (Major Weaknesses 1 and Question 1)**
>
> We appreciate the reviewer’s insightful observation. In **V2X-UniPool**, the infrastructure does not generate individual trajectories or control commands for each vehicle. Instead, it **broadcasts structured and language-based knowledge packets** describing the shared environment (e.g., dynamic agents, occluded risks, or road conditions). Each vehicle then interprets and integrates this information with its own local perception to support autonomous planning. This design makes the system far more **scalable and deployable in real-world V2X infrastructure**, and fully preserve vehicle-side autonomy.
>
> We would also like to clarify that **V2X-UniPool is not an infrastructure-side planner but a language-based perception enhancement layer**. Unlike **V2X-VLM**, which directly predicts waypoints or trajectories through multimodal fusion, UniPool focuses on **extending and enriching perception** via structured knowledge. All downstream vehicle-side planners in our experiments are **frozen pre-trained models** , allowing us to isolate and quantify the incremental gain brought by UniPool’s contextual reasoning rather than re-train the entire stack.
>
> Empirically, **V2X-UniPool consistently improves perception-based planning performance**. Averaged over all backbones, it reduces planning error (L2/ADE) by **4.7 %**, with up to **8.2 %** improvement on Qwen-32B. The mean collision rate drops by **68 %**, reaching a **73 %** reduction on Llama-3.2, and communication cost decreases by **over 80 %** compared with sota.

---

### Official Review · Reviewer_4czq · 2025-11-01

**Soundness:** 3
**Presentation:** 3
**Contribution:** 2
**Rating:** 4
**Confidence:** 4

**Summary:**

This paper presents V2X-UniPool. The core idea is to establish a "V2X as a Knowledge Service" model, where roadside infrastructure (RSUs) acts as a centralized processor. It fuses raw multimodal sensor data (LiDAR, cameras) into a structured, language-based "Knowledge Pool." Vehicles (users) then query this pool to retrieve a snapshot of their environment, which is fused with their local perception for final decision-making by a language model. The authors claim this approach addresses key V2X challenges like heterogeneity and high communication costs. Experiments on the DAIR-V2X dataset show an impressive reduction in communication overhead (>80%) and competitive planning accuracy.

**Strengths:**

- The concept of a centralized knowledge pool is innovative and well-motivated. It effectively abstracts away sensor heterogeneity and provides an interpretable interface for vehicles.
- The paper provides thorough experiments and ablation studies. The reported performance is strong.
- The methodology is described in detail with clear components for knowledge translation, pool construction (static/dynamic), and RAG-based retrieval.

**Weaknesses:**

- The paper's core motivation is that single-vehicle perception is limited and occluded, leading drivers/AI to blindly trust incomplete information and cause accidents. However, the proposed solution replaces one form of blind trust with another. It shifts trust from the vehicle's own sensors to the centralized V2X system. For example,  a person can maliciously place fake "construction ahead" sign that could fool the RSU's vision model, polluting the entire Knowledge Pool. This would cause all connected vehicles in the area to perform unnecessary and potentially dangerous maneuvers.

- The fusion process D = LM(Fuse(V, E)) is a black box. The paper does not test or explain how the system behaves when the local perception V (e.g., "I see a clear road") directly conflicts with the V2X knowledge E (e.g., "Infrastructure reports an obstacle").

- The proposed system hinges on a massive deployment of computationally powerful RSUs across entire cities.

**Questions:**

- How does the Fuse(V, E) function and the subsequent LLM reasoning handle fundamental conflicts between local perception V and infrastructure knowledge E?

- How does the whole system handle adversarial attacks to the V2X system?

- Could you discuss the estimated cost and infrastructure requirements for deploying V2X-UniPool at a city scale?

---

> ### Author Response · Authors · 2025-11-13
> **Response to Reviewer’s Concern on Trust and Decision Authority**
>
> We sincerely thank the reviewer for raising this important and insightful concern regarding the potential “trust shift” in **V2X-UniPool**. This question touches the core philosophy of our framework, and we are glad to clarify the following points:
>
> #### **1. System Positioning**
>
> **V2X-UniPool is not a centralized decision-making system.**  It does not override or dictate the ego vehicle’s planning or control policy. Instead, it functions as a language-based situational augmentation layer that provides semantic context to enrich perception and reasoning. The ego vehicle remains the sole decision-maker, while V2X-UniPool only extends and enhances its perceptual horizon through interpretable, language-based context knowledge.
>
> #### **2. Meaning of $\text{Fuse}(V, E)$**
>
> The fusion function $\text{Fuse}(V, E)$ performs **semantic alignment**, not policy execution. It integrates local perception \(V\) and retrieved environmental knowledge \(E\) into structured, language-based scene descriptions.  These descriptions support reasoning and situational awareness but never directly trigger control actions. Hence, the model acts as a perceptual knowledge interface rather than a decision policy.
>
> #### **3. Addressing the “Trust Shift” Concern**
>
> Each knowledge entry in the pool is associated with metadata (timestamp, source ID, and confidence score). During reasoning, the ego vehicle can filter, down-weight, or ignore inconsistent or low-confidence information. Thus, **V2X-UniPool does not replace trust**; instead, it enables transparent, interpretable, and uncertainty-aware collaboration while preserving full autonomy on the vehicle side.
>
> If there are any further questions or clarifications needed, please feel free to discuss.

---

### Meta-Review · Area_Chair_Ew8m · 2026-01-07

**Summary:**

Reviewers raise concerns regarding the actual usage and performance of this proposed method. Besides, I also find that this work lacks of closed-loop experiments, which is crucial to evaluate driving performance.

Due to the common concerns regarding the performance and lack of closed-loop experiments, I recommend rejection.

**Reviewer Concerns:**

Technical concerns are solved.

However, reviewer 4czq's concerns regrading the protocol and reviewer 2oTn's concerns regarding performance are not solved.

**Reviewer Scores:**

Reviewer 4czq and 2oTn  might keep their negative scores since the raised concerns might not be fully solved.

Reviewer 3myK  might keep his positive score.

---

### Decision · Program_Chairs · 2026-01-26

Reject